# Addressable nanoantennas with cleared hotspots for single-molecule detection on a portable smartphone microscope

Kateryna Trofymchuk [1,2,7✉], Viktorija Glembockyte [1,7✉], Lennart Grabenhorst [1], Florian Steiner [1], Carolin Vietz [2], Cindy Close[1], Martina Pfeiffer[1], Lars Richter [2], Max L. Schütte[2], Florian Selbach [1], Renukka Yaadav [1], Jonas Zähringer [1], Qingshan Wei [3], Aydogan Ozcan[4], Birka Lalkens[5], Guillermo P. Acuna [6✉] & Philip Tinnefeld [1✉]

The advent of highly sensitive photodetectors and the development of photostabilization strategies made detecting the fluorescence of single molecules a routine task in many labs around the world. However, to this day, this process requires cost-intensive optical instruments due to the truly nanoscopic signal of a single emitter. Simplifying single-molecule detection would enable many exciting applications, e.g., in point-of-care diagnostic settings, where costly equipment would be prohibitive. Here, we introduce addressable NanoAntennas with Cleared HOtSpots (NACHOS) that are scaffolded by DNA origami nanostructures and can be specifically tailored for the incorporation of bioassays. Single emitters placed in NACHOS emit up to 461-fold (average of 89 ± 7-fold) brighter enabling their detection with a customary smartphone camera and an 8-US-dollar objective lens. To prove the applicability of our system, we built a portable, battery-powered smartphone microscope and successfully carried out an exemplary single-molecule detection assay for DNA specific to antibiotic-resistant *Klebsiella pneumonia* on the road.

[1] Department of Chemistry and Center for NanoScience, Ludwig-Maximilians-Universität München, München, Germany. [2] Institute for Physical and Theoretical Chemistry - NanoBioScience and Braunschweig Integrated Centre of Systems Biology (BRICS), Technische Universität Braunschweig, Braunschweig, Germany. [3] Department of Chemical and Biomolecular Engineering, North Carolina State University, Raleigh, NC, USA. [4] Electrical & Computer Engineering Department, Bioengineering Department, California NanoSystems Institute (CNSI), and Department of Surgery, University of California, Los Angeles, CA, USA. [5] Institut für Halbleitertechnik, Laboratory for Emerging Nanometrology LENA, TU Braunschweig, Langer Kamp 6a/b, Braunschweig, Germany. [6] Département de Physique - Photonic Nanosystems, Université de Fribourg - Faculté des Sciences et Médicine, Fribourg, Switzerland. [7] These authors contributed equally: Kateryna Trofymchuk and Viktorija Glembockyte. ✉email: kateryna.trofymchuk@cup.lmu.de; viktorija.glembockyte@cup.lmu.de; guillermo.acuna@unifr.ch; philip.tinnefeld@cup.lmu.de

Early detection of disease biomarkers generally requires high sensitivity enabled by molecular amplification mechanisms[1–5] or physical signal enhancement of commonly used fluorescence signals[6–9]. Physical fluorescence signal enhancement could enable sensitivity improvement, detection of single molecules on cost-effective and mobile devices and therefore help to distinguish specific signals against an unavoidable background of impurities even in low-resource settings. Fluorescence from emitters such as fluorescent dyes can be enhanced using plasmonic nanoantennas[10–12], and the challenge of placing quantum emitters in their hotspots was overcome using DNA origami as constructing material[13,14]. The immense requirements for small, defined and rigid gaps between the gold or silver nanoparticles forming the gap in the nanoantenna aggravated the usability of the space between the nanoparticles for a biosensing assay. While it was demonstrated that incorporation of a fluorescence quenched hairpin in a nanoantenna hotspot allowed for the specific detection of DNA specific to Zika virus, the limited accessibility of the hotspot and the steric constraints imposed by the DNA origami nanopillar, the capturing strands and the nanoparticles only allowed for the binding of a single nanoparticle (monomer antenna) strongly reducing the achievable enhancement values (average of 7.3)[15]. These moderate fluorescence enhancement values were not sufficient for detecting single fluorescence molecules with low numerical aperture(NA) optics. For example, our previous work on benchmarking the sensitivity of smartphone-based detection systems suggested that a signal equivalent to at least 16 single emitters is required for detection on a smartphone-based low-NA microscope[16]. Therefore, a diagnostic single-molecule assay fully exploiting the signal amplification potential of DNA origami nanoantennas has not been presented to date and remained highly desirable to enable detection of single molecules with affordable low-NA optics.

In this work, we introduce NanoAntennas with Cleared HOtSpots (NACHOS) that enable high fluorescence signal amplification and are fully addressable, i.e., new analytes can be introduced into the confined regions of dimer nanoantennas. We use these NACHOS for a single-molecule diagnostic assay on a portable and inexpensive smartphone microscope.

## Results

### Design and fluorescence enhancement of NACHOS.
A novel three-dimensional DNA origami structure was designed (Fig. 1a) and folded from an M13mp18-derived scaffold strand and complementary staple strands (Supplementary Tables 1–3). The NACHOS origami design uses two pillars to attach silver nanoparticles and creates the plasmonic hotspot at the bifurcation in the gap between the two pillars and the nanoparticles (see DNA origami sketches in Fig. 1a and full NACHOS structure in Fig. 1b and Fig. 1c). Thus, the space of the hotspot, i.e., between the nanoparticles is left free for placing baits and for binding targets as needed for nucleic acid bioassays. For immobilization, the DNA origami structure is equipped with a rigid cross-like shaped base (approximately 35 nm by 33 nm, Supplementary Figs. 1 and 2) that contains six biotin-modified staple strands (Supplementary Table 3) used for immobilization on BSA-biotin coated coverslips via biotin-NeutrAvidin interactions (Fig. 1b). The two pillars of the DNA origami structure (total height ~83 nm) each contain six protruding staple strands ($A_{20}$, Supplementary Table 3) which provide anchor points for binding DNA ($T_{20}$)-functionalized 100 nm silver nanoparticles (Fig. 1b). The estimated distance between the nanoparticles is ~12 nm. A transmission electron microscopy (TEM) image of an exemplary nanoantenna produced via solution synthesis is shown in Fig. 1c (see Materials and Methods section for details on magnetic bead-based solution synthesis). We evaluated the signal amplification that can be achieved in this DNA origami nanoantenna design by incorporating an Alexa Fluor 647-labeled DNA staple strand (Supplementary Table 3) directly into the plasmonic hotspot of the nanoantenna. Single-molecule fluorescence transients of the dye (Fig. 1d, Supplementary Fig. 3) were recorded on a confocal microscope for the DNA origami sample without nanoparticles (orange) as well as for NACHOS containing two 100 nm silver nanoparticles attached to the DNA origami after immobilization on the coverslip (blue, see Materials and Methods section for NACHOS synthesis on the coverslip). Single-step photobleaching in the intensity versus time transients (Fig. 1c) confirms that the detected signal originates from a single fluorescent molecule. Further analysis of single-molecule transients demonstrates that the signal-to-background ratio (SBR) could be significantly improved by the nanoantenna ($361 \pm 35$) when compared to the reference structure ($7.4 \pm 0.9$). The fluorescence enhancement obtained for each nanoantenna was calculated by comparing the intensity of Alexa Fluor 647 in the NACHOS to the mean intensity of Alexa Fluor 647 in the reference structure without nanoparticles. Fluorescence enhancement values of up to 417-fold (average of $74 \pm 3$-fold) could be achieved in the new nanoantenna design (Fig. 1e). The broad fluorescence enhancement distribution reflects some heterogeneity with regard to nanoparticle size, shape and orientation, and also includes a subpopulation of monomer nanoantennas. Care was taken that all fluorescent molecules incorporated in the DNA origami nanoantennas were included in the analysis to obtain a representative distribution of fluorescence enhancement values in Fig. 1e. Most importantly, we note that increasing the accessibility of the hotspot region did not compromise the fluorescence enhancement values which are slightly higher than previously reported values for more compact nanoantenna designs[14,17,18].

### Amplified single-molecule detection of DNA with NACHOS.
To utilize the plasmonic hotspot for single-molecule diagnostics we designed a sandwich binding assay capable of detecting a DNA fragment specific to *OXA-48*, which is the gene that codes for carbapenemase and is used for the diagnosis of an antibiotic resistant *Klebsiella pneumoniae* infection[19,20]. Three capture strands specific to the target DNA (Supplementary Table 4) were incorporated, protruding directly into the plasmonic hotspot of the NACHOS. The rationale of using three capturing strands was to optimize the probability of each DNA origami having binding strands accessible to capture the target[21]. The principle of this assay is illustrated in Fig. 2a: a 17-nt long capture strand is complementary to one half of the 34-nt long target DNA strand. Binding of the target DNA sequence then provides an overhang for the 17-nt long dye-labeled imager strand to be incorporated directly in the plasmonic hotspot where the signal of the reporter dye is amplified by the nanoantenna. In addition, the DNA origami structure is labeled with a single ATTO 542 dye close to the base.

Surface scans before incubation with the target and imager strands show green fluorescent spots that represent single NACHOS (Fig. 2b, Supplementary Fig. 4). After incubating (2 h at 37 °C) the NACHOS with the target DNA sequence (2 nM, Supplementary Table 4) as well as with the Alexa Fluor 647-labeled imager strand (6 nM, Supplementary Table 4), the presence of the target DNA could be detected and quantified by counting the number of colocalized green (ATTO 542) and red (Alexa Fluor 647) spots in confocal fluorescence scans (Fig. 2c, Supplementary Fig. 4). Although 2 h were used for the assay, we note that significant binding of target sequence in the hotspot of NACHOS was already achieved after 15 min of incubation at

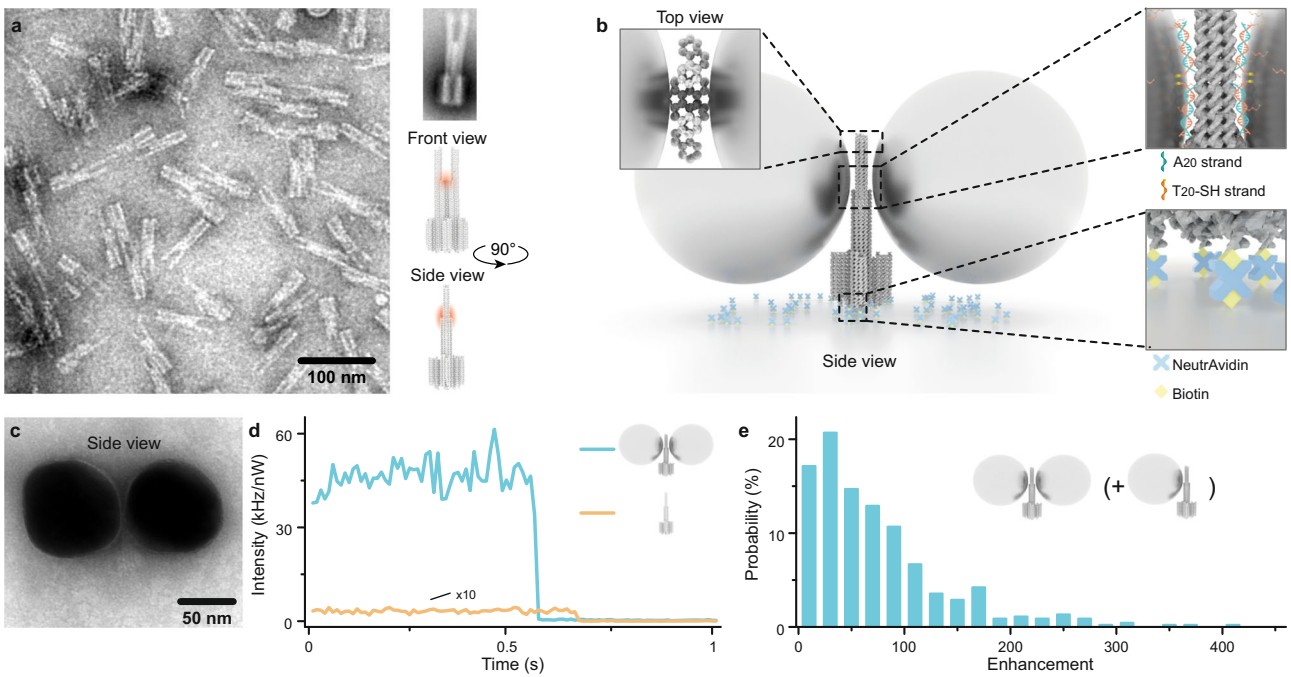

**Fig. 1 Concept of the DNA origami nanoantenna with a cleared hotspot. a** TEM image (left, reproduced at least 3 times) and sketches (right) of the DNA origami structure used for the nanoantenna assembly with the position of the plasmonic hotspot indicated in red. A representative class averaged TEM image of the DNA origami used is shown on the upper right. **b** Schematics of NACHOS assembly: the DNA origami construct is bound to the BSA-biotin coated surface via biotin-NeutrAvidin interactions, thiolated DNA-functionalized 100 nm silver particles are attached to the DNA origami nanoantenna via polyadenine ($A_{20}$) binding strands in the zipper-like geometry to minimize the distance between the origami and the nanoparticles[30]. **c** TEM image of a NACHOS with 100 nm silver nanoparticles (reproduced at least 3 times). **d** Single-molecule fluorescence intensity transients, measured by confocal microscopy, normalized to the same excitation power of a single Alexa Fluor 647 dye incorporated in a DNA origami (orange) and in a DNA origami nanoantenna with two 100 nm silver nanoparticles (blue) excited at 639 nm **e**. Fluorescence enhancement distribution of Alexa Fluor 647 measured in NACHOS with 100 nm silver nanoparticles. A total number of 164 and 449 single molecules in the reference (more examples are provided in Supplementary Fig. 3) and NACHOS structures were analyzed, respectively.

37 °C (Supplementary Fig. 5). When the nanoantennas were incubated with the imager strand only (Fig. 2d, f, and Supplementary Fig. 4), very few co-localized spots were observed. This control demonstrated a low fraction (~2.5%) of false positive signals. Incubation of NACHOS with 34-nt long target sequence containing 1-nt, 2-nt and 3-nt mismatches in the target region led to a drop in the number of co-localized spots (Supplementary Fig. 6), indicating a certain degree of selectivity in this assay, which potentially can be further improved by optimizing the sequence and length of the DNA capture strand.

Next, we studied the fluorescence enhancement that could be achieved in this single-molecule DNA diagnostics assay (Fig. 2e). Fluorescence enhancement values were calculated by comparing the intensity of Alexa Fluor 647 in NACHOS that contained only one dye incorporated in the hotspot (i.e., displayed single-step bleaching events in fluorescence transients) to the intensity of single Alexa Fluor 647 dyes incorporated in the reference structure without nanoparticles. As shown in Fig. 2e, fluorescence enhancement values of up to 461-fold (average $89 \pm 7$-fold) could be achieved representing more than an order of magnitude improvement compared to previous DNA nanoantennas specific to Zika virus[15]. One major advantage of using NACHOS for the sandwich binding assay is that only the signal originating from the specific binding to the target sequence in the zeptoliter volume of the nanoantenna hotspot is amplified. In contrast, any signal originating from non-specific binding of the imager strand to the DNA origami scaffold or the surface of the glass coverslip is not amplified. The clear differentiation between single-molecule emission amplified by the nanoantenna and the one observed from single fluorescent molecules is illustrated in the inset of Fig. 2e.

We quantified the efficiency of the sandwich binding assay in the reference DNA origami structure without nanoparticles as well as in NACHOS containing 100 nm silver nanoparticles by calculating the fraction of DNA origami structures containing the target and imager (% colocalization of green and red spots, Fig. 2f). Binding efficiencies of 66% and 84% were measured in NACHOS (light blue) and in the reference structures (orange), respectively, confirming that the hotspot accessibility for the target DNA sequence is not significantly compromised by attaching two 100 nm silver nanoparticles. We note that ~10 % higher imager binding yield was observed for the reference structure in the presence as well as in the absence of the target strand, which we attribute to higher non-specific sticking of the imager to the reference structure. We hypothesize this non-specific sticking is related to the single-stranded DNA for nanoparticle binding as unspecific binding is reduced after incorporation of two silver nanoparticles in the full nanoantenna construct (Fig. 2f).

To quantify the number of target molecules incorporated in each nanoantenna hotspot, we performed a single-molecule fluorescence photobleaching analysis (Fig. 2g) which allowed us to determine the number of Alexa Fluor 647 imager strands per DNA origami structure by counting the photobleaching steps in single-molecule fluorescence transients (Supplementary Fig. 7). The majority (~60%) of NACHOS contained one imager strand incorporated in the hotspot, one third of nanoantennas contained two imager strands, while three imager strands were observed in ~8–11% of single-molecule transients. The distribution of bleaching steps obtained for NACHOS as well as for the reference structures (Fig. 2g) further supports the observation that the

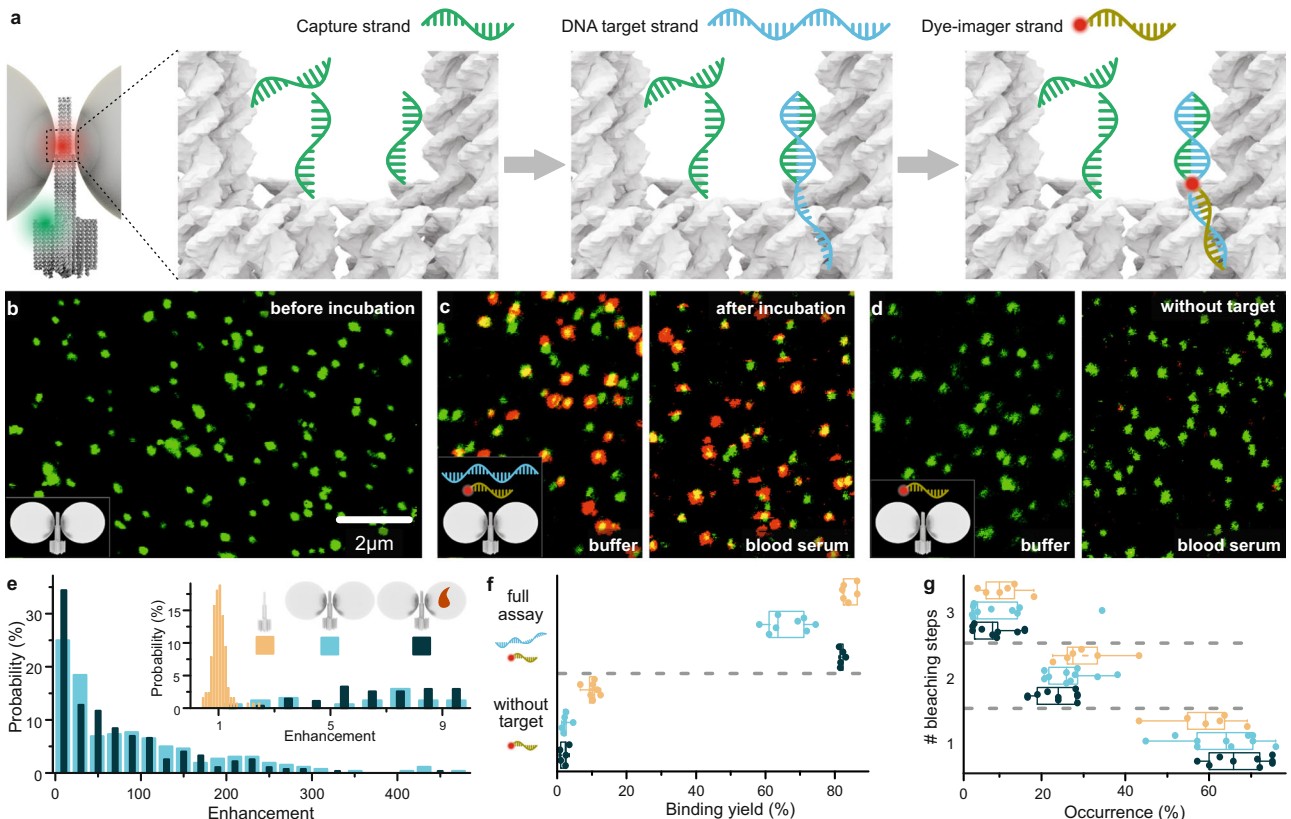

**Fig. 2 Single-molecule diagnostic assay with NACHOS. a** Sketch of NACHOS with three capture strands in the hotspot and a green reference dye (ATTO 542) for labeling of the DNA origami. Capture strands are placed in the NACHOS hotspot region. Upon incubation, they hybridize with DNA target strands specific to *Klebsiella pneumonia*, exposing a specific, 17-nt long region for the hybridization with the imager strand labeled with Alexa Fluor 647. **b** Confocal fluorescence image of the NACHOS before incubation with DNA target and imager strands. **c** Confocal fluorescence images after incubation with DNA target (2 nM) and imager strands (6 nM) in buffer solution (left) and blood serum (right), scale same as in panel **b**; **d** Confocal fluorescence image of the NACHOS after incubation with only the DNA imager strand (6 nM) in buffer solution (left) and blood serum (right), scale same as in panel b. The scans in panels b-d are representative for at least 20 images. **e** Fluorescence enhancement histogram of the sandwich assay in NACHOS measured in buffer solution (light blue) as well as in blood serum (dark blue). The inset includes a zoom in into the enhancement histogram overlaid with an enhancement histogram obtained for non-enhanced single Alexa Fluor 647 dyes (orange). Between 127 and 273 NACHOS and reference structures were analyzed in buffer solution and blood serum, respectively. **f** Binding yield obtained for the full sandwich assay (2 nM target and 6 nM imager strands) and for the control experiment (imager strand only) without nanoparticles (orange) as well as with nanoparticles in buffer solution (light blue) and in blood serum (dark blue). At least 546 spots were analyzed out of at least 5 different areas for each sample. Alexa Fluor 647 was excited at 639 nm and ATTO 542 at 532 nm. **g** Distribution of fluorescence bleaching steps observed in fluorescence transients for NACHOS in buffer solution and in blood serum and reference structures. Over 240 structures from at least 6 different areas per sample were analyzed. The box plots in panels **f** and **g** show the 25/75 percentiles and the whisker represents the 1.5*IQR (inter quartile range) length, the center lines represent the average values.

presence of silver nanoparticles does not obstruct the hotspot accessibility for the DNA target.

**Single-molecule detection in human blood serum**. To demonstrate that NACHOS can still function in complex biological fluids that compromise many diagnostic assays, we have also performed the sandwich detection assay described above in human blood serum spiked with the target DNA sequence specific to the *OXA-48* gene. The serum was first heat-inactivated and then enriched with 2 nM target DNA sequence as well as 6 nM Alexa Fluor 647 imager strand. The fully assembled NACHOS were then incubated in the serum mixture for 2 h at 37 °C. Fluorescence scans of the NACHOS after incubation with serum and target DNA sequence are included in Fig. 2c, d (as well as Supplementary Fig. 8). Almost identical fluorescence enhancement values (Fig. 2e), target binding efficiencies (Fig. 2f) and number of single-molecule photobleaching steps (Fig. 2g) were obtained for reference and NACHOS samples in highly purified buffer (light blue) and serum (dark blue) conditions

confirming that neither the stability of NACHOS nor the performance of the sandwich assay in NACHOS are compromised. On the contrary, fluorescence enhancement values reaching 457-fold (average of 70 ± 4) could be achieved for the DNA detection assay in target spiked human serum. These findings proof the robustness of NACHOS under realistic assay conditions and provide an important stepping stone towards diagnostic applications.

**Single-molecule detection on a portable microscope using NACHOS**. Recently, the detection of only 10–16 ATTO 542 molecules was demonstrated using a simple table top setup with a monochrome smartphone camera as detector and a consumer product lens for light collection[16]. This inspired us that single-molecule detection might be possible on a portable smartphone microscope with non-specialized low-NA optics[2,22–24] (see Fig. 3a, b). The microscope uses the monochrome camera of a Huawei P20 smartphone for detection, data processing and interfacing and a battery-driven 638 nm excitation laser with

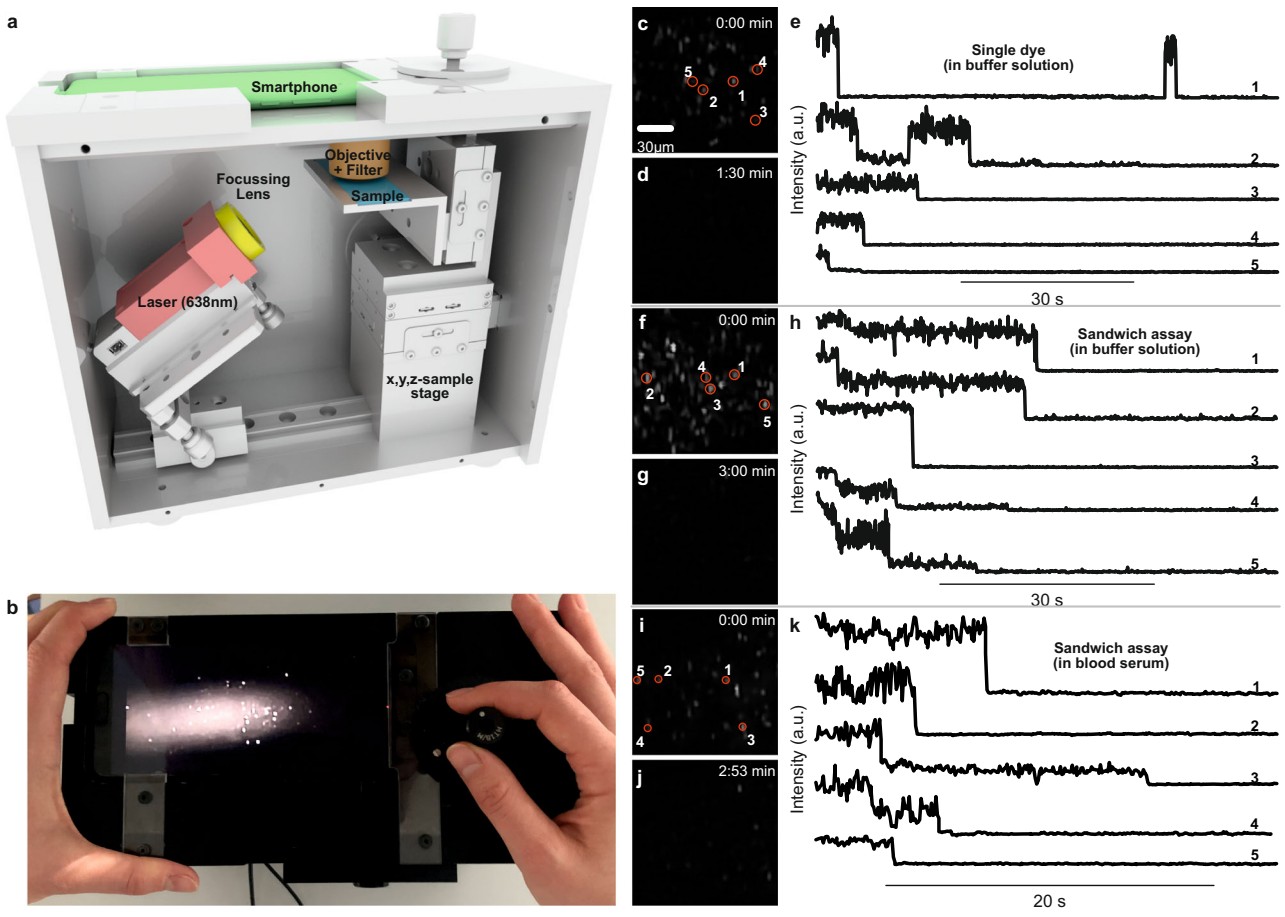

**Fig. 3 Single-molecule detection on a portable smartphone microscope. a** Sketch of the portable smartphone microscope with the battery driven 638 nm laser (red), the focusing lens ($f = 5$ cm) (yellow), the microscope coverslip with the sample (blue), the objective lens and the emission filter (brown), and the smartphone monochrome camera as detector (green). **b** Top view photograph of the portable smartphone microscope. **c** Background corrected fluorescence image of NACHOS with 100 nm silver nanoparticles and a single Alexa Fluor 647 dye. **d** Fluorescence image as in **c** after illumination for 1:30 min. **e** Exemplary fluorescence transients of a single Alexa Fluor 647 in NACHOS measured on the portable microscope setup. Single bleaching steps of dyes and long-time blinking events are visible. **f** Background corrected fluorescence image of NACHOS equipped with a sandwich assay with 100 nm silver nanoparticles and Alexa Fluor 647 imager strands. **g** Fluorescence image as in **f** after illumination of the area for 3:00 min. **h** Exemplary fluorescence transients of Alexa Fluor 647 in a three-capture-strand DNA origami nanoantenna measured on the portable smartphone microscope. **i** Background corrected fluorescence image of NACHOS equipped with the sandwich assay with 100 nm silver nanoparticles and Alexa Fluor 647 imager strands after incubation in blood serum. **j** Fluorescence image as in **i** after illumination of the area for 2:53 min. **k** Exemplary fluorescence transients of Alexa Fluor 647 in a three-capture-strand NACHOS measured on the portable smartphone microscope. Fluorescence transients with one, two, and three bleaching steps (analogous to single-molecule confocal measurements) were observed. The movies represented in the panels **c**, **d**, **f**, **g**, **i** and **j** were reproduced at least 5 times. Three movies for each measurement are provided in the Supplementary Movies. The fluorescence transients shown in panels **e**, **h** and **k** were extracted from a single movie.

180 mW output power. The excitation laser (red in Fig. 3a) is focused on the sample plane at approximately 45° using a lens with a focal length of 5 cm to illuminate an elliptical area of ~150 × 200 µm². Fluorescence emission is collected and collimated with a consumer product lens (NA = 0.25, 8 US$, yielding a resolution of ~1.2 µm in the red wavelength range), bandpass filtered and focused onto the smartphone detector using the internal lens in the infinite focal distance mode. A discussion of the total price of the components used in the prototype smartphone microscope (sum of ~ 4200 €) can be found in Supplementary Note 1. We envision that the price can be reduced in the case of upscaling production (<1000 €). Importantly, the affordable microscope does not imply expensive sample preparation. The single-molecule nature of measurements requires substantial dilution of the DNA origami samples and DNA functionalized nanoparticles yielding an estimated price per NACHOS coverslip preparation of below 5 € (Supplementary Note 2).

First, we prepared NACHOS with 100 nm silver nanoparticles and a single Alexa Fluor 647 dye in the hotspot. Considering the low resolution of the smartphone microscope, the concentration of NACHOS on the surface was adjusted to a reasonably low density to ensure that only one nanoantenna is present per diffraction limited spot (see Materials and Methods section). To improve the photostability of Alexa Fluor 647 and demonstrate single-molecule bleaching steps, the measurements were carried out in a reducing and oxidizing system (ROXS)[25,26] with enzymatic oxygen removal. Upon illumination, multiple bright spots were observed on the smartphone screen (Fig. 3c). In the movies recorded with 80 ms per frame, slow single-molecule blinking and bleaching (Supplementary Fig. 9) was observed (see Supplementary Movies 1–3) as indicated by the disappearance of spots over time (compare Fig. 3c, d). Extracted fluorescence transients (spots from one movie) are shown in Fig. 3e, demonstrating typical single-molecule behavior with

blinking and single-step bleaching events. These transients represent the first examples of single-molecule fluorescence detection with a portable smartphone microscope and non-dedicated optics bringing single-molecule detection a step closer to point-of-care settings. The signal-to-background ratio (SBR) and the signal-to-noise ratio (SNR) of the transients on the smartphone microscope are determined to be $25 \pm 2$ and $3.8 \pm 0.2$, respectively. Examples of fast blinking of single ATTO 647N dyes in the hotspot of NACHOS with 100 nm silver nanoparticles can be found in Supplementary Movie 4 and Supplementary Fig. 10.

Next, we tested whether the portable smartphone microscope could also be used for the detection of single DNA molecules in analogy to the sandwich assay discussed in Fig. 2. The sandwich assay with three capture strands for the detection of the resistance gene OXA-48 imaged with the portable smartphone microscope is shown in Fig. 3f. All fluorescence spots acquired on the smartphone camera were photobleached after 3 min of movie recording (see Supplementary Movies 5–7). The extracted transients (Fig. 3h) exhibit bleaching of the imager strands with 1–3 bleaching steps in accordance with the single-molecule fluorescence transients acquired on the confocal microscope shown in Supplementary Fig. 7. More examples of extracted transients for the sandwich assay with three binding strands in the NACHOS hotspot are included in Supplementary Fig. 12. In control measurements under identical conditions leaving out the nanoparticles, no signal could be detected. As a further control, we incubated the coverslips with silver nanoparticles only. A few dim spots that did not disappear after long illumination are ascribed to scattering from silver nanoparticle aggregates (Supplementary Fig. 11). These results confirm that single-molecule detection of disease-specific DNA can also be performed on our portable smartphone microscope omitting the need for advanced and expensive microscopes. Finally, the DNA detection assay after incubation with human blood serum was also measured on the portable smartphone microscope. Images at the beginning as well as at the end of the movie and exemplary fluorescence transients are shown in Fig. 3i, j, k. The results are almost identical to the measurements in purified buffer solution (Fig. 3f–h) with a decreasing number of isolated fluorescent spots detected on the camera (Fig. 3i, j) due to photobleaching. In a similar way the fluorescent transients (Fig. 3k) show clear single, double and triple bleaching steps with no difference visible between the purified buffer and the blood serum assays. More example movies and transients for the measurements of the sandwich assay inside the NACHOS are shown in Supplementary Movies 8–10 and Supplementary Fig. 13. The photobleaching analysis for the transients from the movie taken on the smartphone microscope is shown in Supplementary Fig. 14 and yields similar distributions for single, double and triple photobleaching steps as compared to the data shown in Fig. 2g, highlighting the ability of the smartphone microscope in combination with NACHOS to provide analytical power comparable to conventional single-molecule microscopy tools.

Self-assembled nanoantennas with a cleared and addressable hotspot represent an inexpensive and versatile platform for fluorescence signal enhancement assays. Single fluorescent molecules immobilized in the hotspot of these newly designed nanoantennas yield higher fluorescence enhancement values than previous approaches with hotspots blocked by the DNA origami nanostructure. NACHOS are robust (see Supplementary Fig. 15 for single-molecule data of a similar sample measured over 13 weeks), stable in complex biological fluids such as human serum, and importantly, the accessibility of the hotspot for target DNA molecules and imagers is not impaired despite the constricted dimensions. A single-molecule sandwich assay with three capturing strands shows equally high fluorescence

enhancement as direct incorporation of a single fluorescent dye in the hotspot and enables single-molecule detection with amplified signal that facilitates discrimination of single-molecule binding events against an unavoidable background of single-molecule impurities (Fig. 2e inset). The demonstration of single-molecule assays on a simple battery-operated smartphone microscope makes DNA origami nanoantennas a stepping-stone for democratizing single-molecule detection with cost-effective and mobile devices relevant for point-of-care applications.

## Methods

**DNA origami**. DNA origami structures were designed in caDNAno2[27] and assembled and purified using protocols inspired by Wagenbauer et al.[28]. Briefly, 25 µL of p8064 scaffold (produced in-house) at 100 nM were mixed with 18 µL of unmodified staples pooled from 100 µM original concentration and 2 µL of modified staples, pooled from 100 µM original concentration. All staples were purchased from Eurofins Genomics GmbH (Germany) - for the exact sequences see Supplementary Table 2. 5 µL of folding buffer (200 mM $MgCl_2$, 50 mM Tris, 50 mM NaCl, 10 mM EDTA) were added and the mixture was subjected to a thermal annealing ramp (see Supplementary Table 1). Samples were purified using 100 kDa MWCO Amicon Ultra filters (Merck KGaA, Germany) with 4 washing steps with a lower ionic strength buffer (5 mM $MgCl_2$, 5 mM Tris, 5 mM NaCl, 1 mM EDTA) for 8 mins at 8 krcf, 20 °C.

**Functionalization of silver nanoparticles**. 100 nm silver nanoparticles (100 nm BioPure Silver Nanospheres (Citrate), nanoComposix, USA) were functionalized with $T_{20}$ single-stranded DNA oligonucleotides with a thiol modification at the 3'-end (Ella Biotech GmbH, Germany)[15]. Briefly, 2 mL of 0.025 mg/mL nanoparticle solution in ultra pure water was heated to 40 °C under permanent stirring. 20 µL of 10 % Tween 20 and 20 µL of a potassium phosphate buffer (4:5 mixture of 1 M monobasic and dibasic potassium phosphate, Sigma Aldrich) were added as well as 10 µL of a 2 nmol thiol-modified single-stranded DNA solution (5'-$T_{20}$-SH-3') and incubated for 1 h at 40 °C. A salting procedure was then carried out by adding 1× PBS buffer containing 3.3 M NaCl stepwise over 45 min to the heated and stirred solution, until a final concentration of 750 mM NaCl was reached. Afterwards, the particles were mixed 1:1 with 1× PBS 10 mM NaCl, 2.11 mM P8709 buffer (Sigma Aldrich, USA), 2.89 mM P8584 buffer (Sigma Aldrich, USA), 0.01 % Tween® 20 and 1 mM EDTA. To remove the excess thiolated single-stranded DNA, the solution was centrifuged for 15 min at 2.8 krcf and 20 °C. A pellet was formed in which the particles were concentrated. The supernatant was discarded, and the washing step was repeated six more times. After functionalization of the silver nanoparticles were diluted in 1× TE buffer (10 mM Tris, 1 mM EDTA) containing 750 mM NaCl to reach the final extinction of 0.05 (0.1 mm path length) at the extinction maxima on a UV-Vis spectrometer (Nanodrop 2000, Thermo Fisher Scientific, USA).

**Solution synthesis of DNA origami nanoantennas for TEM imaging**. To obtain DNA origami nanoantennas in solution, the structures were initially assembled on streptavidin-coated magnetic beads (Dynabeads™ MyOne™ Streptavidin C1, 1 µm diameter, 10 mg/mL, Thermo Fischer Scientific, USA). Preparation of magnetic beads: 3.0 µL of bead stock solution (~20–30 ×10[6] beads) were washed three times with 50 µL 1× B&W buffer (0.5 mM EDTA, 5 mM Tris-HCl (pH = 8), 1 M NaCl, 0.001 % v/v Tween® 20). After removing the supernatant, the beads were diluted in 6.0 µL 1× B&W and incubated with 6.0 µL of 4 µM biotinylated ssDNA (mag1, Supplementary Table 5) for 20 min at room temperature. The functionalized beads were purified from excess of ssDNA by placing the tube on a magnet and discarding the supernatant. The beads were redispersed in 50 µL 1× B&W and washed with 1× B&W buffer (3× 50 µL). Immobilization of DNA Origami on Magnetic Beads: DNA origami (100 µL, 200 pM in 1× B&W buffer) with three ssDNA overhang strands on a bottom partially complementary to the sequence on the magnetic beads (mag2, Supplementary Table 5) were incubated together for 2 h at 37 °C under gentle shaking (450 rpm, Eppendorf ThermoMixer® C, Eppendorf AG, Germany). Unbound DNA origami was removed by placing the tube on a magnet and discarding the supernatant. The beads were redispersed in 50 µL 1× B&W and washed with 1× B&W buffer (5× 50 µL). Binding of 100 nm silver nanoparticles: Nanoantennas were fabricated on magnetic beads by hybridizing with DNA functionalized (5'-$T_{20}$-SH-3') 100 nm silver nanoparticles to the DNA origamis. For this the supernatant of the with DNA origami coated beads was removed and incubated with 100 µL of 100 nm silver nanoparticles solution using an excess of five nanoparticles per binding site. During the first three hours of incubation, the solution was mixed every 30 min by gentle pipetting. After overnight incubation at room temperature, the excess of nanoparticles was removed by placing the tube on a magnet and discarding the supernatant. The beads were re-dissolved in 50 µL 1× B&W and washed with 1× B&W buffer (5× 50 µL). Cleavage of the assembled structures: Nanoantennas were cleaved from the beads surface by performing a toehold-mediated strand displacement reaction. For cleavage, the supernatant of

the bead solution was removed and nanoantennas coated beads were incubated with 20 µL 10 µM of the displacement strand (mag3, Supplementary Table 5) for 4 h at room temperature. Unbound DNA origami nanoantennas were recovered for further use by placing the tube on a magnet.

**Transmision electron microscopy (TEM) measurements**. TEM grids (Formvar/carbon, 400 mesh, Cu, TedPella, Inc., USA) were Ar-plasma cleaned and incubated for 60 s with DNA origami sample (5 µL, ~ 2–10 nM). Grids were washed with 2 % uranyl formate solution (5 µL, ~ 2–10 nM). Grids were washed with 2 % uranyl formate solution (5 µL) and incubated again afterwards again 4 s with 2% uranyl formate solution (5 µL) for staining. TEM imaging were performed on a JEM-1100 microscope (JEOL GmbH, Japan) with an acceleration voltage of 80 kV.

**Sample preparation on the coverslip for single-molecule confocal measurements**. Microscope coverslips of 24 mm × 60 mm size and 170 µm thickness (Carl Roth GmbH, Germany) were cleaned with UV-Ozone cleaner (PSD-UV4, Novascan Technologies, USA) for 30 min at 100 °C. Adhesive SecureSeal™ Hybridization Chambers (2.6 mm depth, Grace Bio-Labs, USA) were glued on the clean coverslips. The created wells were washed three times with PBS buffer and then incubated with BSA-biotin (0.5 mg/mL, Sigma-Aldrich, USA) and NeutrAvidin (0.2 mg/mL, Thermo Fisher Scientific, USA). The DNA origami (50–100 pM in 1× TE buffer containing 750 mM NaCl) was immobilized on the biotin-NeutrAvidn surfaces using covalently attached biotin modifications on the six staple strands on the base. Density of DNA origami nanoantennas on the surface suitable for single-molecule measurements was checked on a microscope. The binding of silver nanoparticles was then performed by incubating the surfaces with 100 µL of $T_{20}$-functionalized silver nanoparticles in 1× TE buffer containing 2 M NaCl overnight at room temperature. To prevent the evaporation of samples, samples were kept in a sealed humidity chambers during the incubation. The nanoantennas were then imaged in 1× TE buffer containing 14 mM $MgCl_2$.

**Diagnostic sandwich assay**. To specifically detect the DNA sequence specific to the *OXA-48* gene carrying the antibiotic resistance[19,20], DNA origami nanoantennas were folded containing three specific capture strands (Supplementary Table 4) extruding from the hotspot region of the nanoantenna. After the assembly of the full nanoantenna in the analogous way to the previous section, the samples were incubated with 2 nM target DNA sequence (34 nt) specific to the *OXA-48* gene (Supplementary Table 4) as well as 6 nM Alexa Fluor 647 imager strand (17 nt) labeled with Alexa Fluor 647 (Supplementary Table 4) in 1× TE buffer containing 2 M NaCl. The sample was incubated for at 37 °C for 2 h and the excess of the target and imager strands was removed by washing six times with 1× TE buffer containing 2 M NaCl. The nanoantennas were then imaged in 1× TE buffer containing 14 mM $MgCl_2$.

For the sandwich assay in serum clotted, whole blood, sterile and filtered human blood serum (Human Serum, (from male AB clotted whole blood), USA origin, sterile-filtered, Sigma-Aldrich, USA) was used. Before adding the serum to the NACHOS and reference samples, the serum was heat inactivated by exposing it for 30 min to 56 °C and spiked with 2 nM target DNA, 6 nM imager strand and 2 M NaCl. The fully assembled NACHOS or reference DNA origami structures were incubated with target-spiked blood serum for 2 h at 37 °C and the excess of target and imager strands was removed by washing six times with 1× TE buffer containing 2 M NaCl. NACHOS were then imaged in 1× TE buffer containing 14 mM $MgCl_2$.

**Confocal measurements and data analysis**. Confocal fluorescence measurements were performed using a home-built confocal setup based on an inverted microscope (IX-83, Olympus Corporation, Japan) and a 78 MHz-pulsed supercontinuum white light laser (SuperK Extreme EXW-12, NKT Photonics A/S, Denmark) with selected wavelengths of 532 nm and 639 nm. The wavelengths are selected via an acousto-optically tunable filter (AOTF, SuperK Dual AOTF, NKT Photonics A/S, Denmark). This is controlled by a digital controller (AODS 20160 8 R, Crystal Technology, USA) via a computer software (AODS 20160 Control Panel, Crystal Technology, Inc., USA). A second AOTF (AA.AOTF.ns: TN, AA Opto-Electronic, France) was used to alternate 532 nm and 639 nm wavelengths if required, as well as to further spectrally clean the laser beam. It is controlled via home-made LabVIEW software (National Instruments, USA). A neutral density filter was used to regulate the laser intensity, followed by a linear polarizer and a λ/4 plate to achieve circularly polarized excitation. A dichroic beam splitter (ZT532/640rpc, Chroma Technology, USA) and an immersion oil objective (UPlanSApo 100×, NA = 1.4, WD = 0.12 mm, Olympus Corporation, Japan) were used to focus the excitation laser onto the sample. Micropositioning was performed using a Piezo-Stage (P-517.3CL, E-501.00, Physik Instrumente GmbH&Co. KG, Germany). The excitation powers at 639 nm were set to 200 nW or for 500 nW for the reference samples and to 50 nW for the NACHOS for the recording of the fluorescence transients. These powers were chosen to ensure that the samples are excited in the linear regime and to avoid saturation in the nanoantenna hotspot[29]. For the confocal scans, 2 µW at 532 nm and 2 µW and 500 nW at 639 nm were used for the reference and NACHOS samples, respectively. Emitted light was then collected using the same objective and filtered from the excitation light by the dichroic beam splitter. The

light was later focused on a 50 µm pinhole (Linos AG, Germany) and detected using avalanche photodiodes (SPCM, AQR 14, PerkinElmer, Inc., USA) registered by an TCSPC system (HydraHarp 400, PicoQuant GmbH, Germany) after additional spectral filtering (RazorEdge 647, Semrock Inc., USA for the red channel and BrightLine HC 582/75, Semrock Inc., USA for the green channel). A custom-made LabVIEW software (National Instruments, USA) was used to process the acquired raw data. Background correction was made individually for each transient. The extracted data were analyzed in OriginPro2016.

**Sample preparation for single-molecule measurements on the smartphone microscope**. The geometry of the smartphone-based microscope required samples to be sealed. To this end, microscope cover slides of 22 mm × 22 mm size and 170 µm thickness (Carl Roth GmbH, Germany) were cleaned with UV-Ozone cleaner (PSD-UV4, Novascan Technologies, USA) for 30 min at 100 °C. After this a home-made silicon mask with an opening around 15 mm × 15 mm was glued on a coverslip to create an incubation chamber. Surface functionalization, DNA origami immobilization (5 - 10 pM), nanoantenna formation, and the sandwich sensing assay were performed the same was as described above for the NACHOS assembly on coverslips. To seal the samples, the silicon mask was removed, and a double-sided tape was glued on both sides of the cover slide. Then the cover slides were covered with 76 mm × 26 mm microscope slides (1 mm thickness, Carl Roth GmbH, Germany) which were priory cleaned with UV-Ozone cleaner for 30 min at 100 °C. Due to limited photostability of Alexa Fluor 647, samples containing the sandwich assay were imaged in the presence of ROXS photostabilization system. A reducing and oxidizing buffer system with enzymatic oxygen removal consisting of 90 % buffer A (14 mM $MgCl_2$, 50 mM Tris, 2 mM Trolox/Troloxquinone and 1 % w/v Glucose) and 10 % buffer B (glucose oxidase (1 mg/mL), 0.4 % (v/v) catalase (50 µg/mL), 30 % glycerol, 12.5 mM KCl) was used. After this the chambers were sealed with nail polish and imaged after the drying of the glue.

**Single-molecule measurements and analysis on the smartphone**. Single-molecule measurements on the smartphone were performed using a home-built portable box. The 638 nm laser diode (0638L-11A, Integrated Optics, UAB, Lithuania) with an output power 180 mW that can be driven by various (portable) voltage sources (Power plug, mobile power bank, (rechargeable) batteries) was focused ($f = 50$ mm) in 45° angle onto the sample. The fluorescence of the molecules was collected using an objective lens (NA = 0.25, LS-40166, UCTRONICS, USA) guiding the light to the monochrome camera of the smartphone (P20, Huawei, China) after spectral filtering (BrightLine HC 731/137, Semrock Inc., USA). Movies were recorded via FreeDCam application and analyzed with ImageJ (FIJI) equipped with FFMPEG plugin using a home written macro to convert MP4 format of the acquired movies to a TIFF format and find the single-molecule signals and extract the fluorescence intensity as a function of illumination time. The extracted data were analyzed in OriginPro2016.

**Reporting summary**. Further information on research design is available in the Nature Research Reporting Summary linked to this article.

## Data availability
The raw data acquired in this study are available in a public Zenodo repository (DOI: 10.5281/zenodo.4384169). This includes TEM images, raw and analyzed confocal data, raw movies acquired on the smartphone device, as well as the caDNAno file for the DNA origami nanostructure reported in this work. Further information is available from the authors upon request.

## Code availability
A custom script used to analyze the movies obtained on the smartphone device is available in the Zenodo repository under DOI: 10.5281/zenodo.4384169.

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

## Acknowledgements
The authors thank Vivien Behrendt and Benedikt Hauer (Fraunhofer-Institut für Physikalische Messtechnik IPM, Freiburg, Germany) for cooperation on the assay development and Prof. Tim Liedl/Prof. Joachim Rädler (Ludwig-Maximilians-Universität, Department für Physik, Munich, Germany) for providing access to their facilities especially to the transmission electron microscope. The authors thank Tomas Gisicius for manufacturing the portable smartphone microscope. P.T. gratefully acknowledges financial support from the DFG (INST 86/1904-1 FUGG, excellence clusters NIM and e-conversion), BMBF (Grants POCEMON, 13N14336, and SIBOF, 03VP03891), and the European Union's Horizon 2020 research and innovation program under grant agreement No. 737089 (Chipscope). G.P.A. gratefully acknowledges support by the Swiss National Science Foundation through the National Center of Competence in Research Bio-Inspired Materials and through grant number 200021_184687. V.G. and K.T. acknowledge the support by Humboldt Research Fellowships from the Alexander von Humboldt Foundation. A.O. acknowledges the support of NSF PATHS-UP and HHMI. BL acknowledges funding by the Deutsche Forschungsgemeinschaft (DFG, German Research Foundation) under Germany's Excellence Strategy – EXC-2123 QuantumFrontiers – 390837967 and "Niedersächsisches Vorab" through "Quantum- and Nano-Metrology (QUANOMET)" initiative within the project NL-1.

## Author contributions
P.T., A.O. and G.P.A. conceived the project, L.G. and B.L. developed the DNA origami structure, K.T., V.G. and M.P. optimized the solution synthesis procedure, F.Se. performed the TEM measurements, K.T., V.G., C.C., M.P. and R.Y. developed the sandwich assay and prepared samples, performed and analyzed the measurements on the confocal microscope, C.V., L.R., M.L.S., Q.W., A.O. and G.P.A. worked on an earlier version of the smartphone microscope, K.T., V.G., F.St. and J.Z. constructed the portable smartphone microscope, K.T., V.G. and F.St. performed and analyzed the measurements on the smartphone microscope, K.T., V.G., L.G., F.St. and P.T. wrote the manuscript. All authors have read and aproved the final manuscript.

## Funding

## Competing interests
P.T. and G.P.A. are inventors on an awarded patent of the described bottom-up method for fluorescence enhancement in molecular assays, EP1260316.1, 2012, US20130252825 A1. The remaining authors declare no competing interests.
