## [Peer Review File · Nature Communications]

REVIEWERS' COMMENTS

Reviewer #1 (Remarks to the Author):

The revised version of the manuscript: "Addressable Nanoantennas with Cleared Hotspots for Single-Molecule Detection on a Portable Smartphone Microscope" is much improved and addresses many of my previous concerns. In particular, I appreciate the additional experiments performed to explore the performance of the assay under conditions closer to those that might be encountered in a diagnostic setting. It is shown that the assay achieves similar performance for measurements in blood serum compared to purified buffer, and in the reply to reviewer comments, it is shown that binding of the target still occurs with longer target ssDNA molecules (151nt, vs 34nt), albeit with lower efficiency (43% vs 67%). These new experiments have strengthened this work. I have several remaining concerns.

1. I appreciate the test of specificity of target detection shown in supplementary figure 6. The authors need to discuss in more detail what this means regarding their claims of "diagnostic power" of their assay. To me, the result presented in Supplementary Figure 6 suggests that the assay is not very specific, as 3 nt mismatch still produces as many as 50% of the colocalization spots compared to the full target, which indicates a high false positive rate in the presence of DNA bearing partial complementarity.
2. It took me a few reads to fully appreciate what's new in this work compared to the authors' previous work on this topic. I think the presentation can be improved by explicitly comparing to past work, for example, pointing out that comparing to Ochmann et al. (*Anal. Chem.* 89, 13000), the larger enhancement achieved here made it possible to use a low-NA optics for single-molecule detection.
3. The authors have clarified what they meant by "addressable" in their response letter. However, Fig.1 is not really clear. For example, in panel a, the "gap" between the two pillars is only shown in a tiny inset. In panel b, the gap is NOT shown at all. For panel c, it is not clear which feature the readers should focus on. (Lines 64-65 points to panels b and c to illustrate the "gap" between the pillars). Please consider revising this figure.
4. There are a lot of errors in the SI:
 - a. Supplementary Figure 4 is exactly the same as Supplementary Figure 8, while they are supposed to represent results under completely different conditions.
 - b. SI figure 4 caption, "excitation" is misspelled
 - c. SI table 5 caption, "supplementary" is misspelled
 - d. SI figure 15 is presented in a confusing manner. It seems to be three histograms offset arbitrarily along the horizontal axis?
5. In Supplementary Note 1, the authors claim that "large scale production of the filters and adapted filter size can reduce the price by at least one order of magnitude." I do not understand where this statement comes from. Please provide more reasoning or source. The current commercial filters are already produced at a large scale. Customized coating runs are still much more expensive (for example, as offered by Chroma).
6. Some sentences lack clarity.
 - a. Line 80-83 "Single-step photobleaching confirms that the detected signal originates from a single fluorescent molecule with a signal-to-background ratio (SBR) of 7.4 ± 0.9 and 361 ± 35 and a signal-to-noise ratio (SNR) of 4.5 ± 0.2 and 5.6 ± 0.3 for the reference structures and the NACHOS, respectively." this sentence is difficult to read.
 - b. Fig 1d caption: "Single-molecule fluorescence intensity transients, measured by confocal microscopy, normalized to the same excitation power of a single Alexa Fluor 647 dye". There is no "normalization" from the figure as count rates were reported.
 - c. SI Table 4 caption: "If necessary, some unmodified strands from Supplementary Table 2 and modified strands from Supplementary Table 3 should be left out." I'm not sure what exactly this means.

Reviewer #2 (Remarks to the Author):

This revised manuscript is now addressing properly most of the reviewers concerns. I was already quite positive with the early version, and now that it has improved I can recommend its publication in its present form. The additional concerns remaining (limit of detection, NA for detection optics, bioassay application) fall beyond the scope of this paper as they deserve an in-depth study on their own.

Reviewer #3 (Remarks to the Author):

The revised manuscript addresses the majority of comments by the referees appropriately. Although the manuscript still only provides a proof-of-principle sensing assay, I believe that the technological advances made are very convincing and suitable for Nature Communications. It is particularly nice to see that the DNA origami assay is stable in a more realistic biological fluid. I have one final remark regarding line 131, where the authors mention "Incubation of NACHOS with 34-nt long target sequence containing 1-nt, 2-nt and 3-nt mismatches in the target region also demonstrated the selectivity of this assay for small target sequence variations (Supplementary Figure 6)." This can be mis-interpreted as a claim that the assay is very selective and few-nucleotide mismatches can be distinguished. Looking at the results in the SI however it will be difficult to detect mismatches because the binding yield only reduces by 50% for a 3nt mismatch. This is not too surprising since the capture strand contains 17 nucleotide complementary regions that exhibit a sizeable affinity even with 3 mismatches. I don't believe this will be an issue because in the foreseen application it is highly unlikely that strands with only a few mismatches are present at similar concentrations. I do suggest however to rephrase this sentence because it seems to overclaim the results now.

Point-by-point response to the reviewers' comments

REVIEWERS' COMMENTS

Reviewer #1 (Remarks to the Author):

The revised version of the manuscript: "Addressable Nanoantennas with Cleared Hotspots for Single-Molecule Detection on a Portable Smartphone Microscope" is much improved and addresses many of my previous concerns. In particular, I appreciate the additional experiments performed to explore the performance of the assay under conditions closer to those that might be encountered in a diagnostic setting. It is shown that the assay achieves similar performance for measurements in blood serum compared to purified buffer, and in the reply to reviewer comments, it is shown that binding of the target still occurs with longer target ssDNA molecules (151nt, vs 34nt), albeit with lower efficiency (43% vs 67%). These new experiments have strengthened this work. I have several remaining concerns.

Response: We appreciate the positive feedback from the reviewer and address the remaining concerns point-by-point below.

1. I appreciate the test of specificity of target detection shown in supplementary figure 6. The authors need to discuss in more detail what this means regarding their claims of "diagnostic power" of their assay. To me, the result presented in Supplementary Figure 6 suggests that the assay is not very specific, as 3 nt mismatch still produces as many as 50% of the colocalization spots compared to the full target, which indicates a high false positive rate in the presence of DNA bearing partial complementarity.

Response and action: We fully agree with the reviewer. Certainly, we do not claim that we could achieve a higher selectivity than a comparable assay with nanoantenna structure. As also confirmed by reviewer 3, the assay design is not made for outstanding selectivity but we intended to show that reasonable selectivity as for an assay outside the nanoantenna could be achieved. We adapted our wording to not overclaim selectivity: "Incubation of NACHOS with 34-nt long target sequence containing 1-nt, 2-nt and 3-nt mismatches in the target region led to a drop in the number of co-localized spots (Supplementary Figure 6), indicating a certain degree of selectivity in this assay, which potentially can be further improved by optimizing the sequence and length of the DNA capture strand".

2. It took me a few reads to fully appreciate what's new in this work compared to the authors' previous work on this topic. I think the presentation can be improved by explicitly comparing to past work, for example, pointing out that comparing to Ochmann et al. (Anal. Chem. 89, 13000), the larger enhancement achieved here made it possible to use a low-NA optics for single-molecule detection.

Response and action: We thank the reviewer for her/his suggestion to better emphasize the novelty of our findings. We have already included a comparison of fluorescence enhancement values obtained with the previously reported design (Ochmann et al., Anal. Chem. 89, 13000) and the one reported here in the previous round of revisions. However, to emphasize that this improvement in fluorescence enhancement was absolutely necessary for single-molecule detection with low-NA optics, we have added more details and expanded the explanation in the introductory paragraph: "While it was demonstrated that incorporation of a fluorescence quenched hairpin in a nanoantenna hotspot allowed for the specific detection of DNA specific

to Zika virus, the limited accessibility of the hotspot and the steric constraints imposed by the DNA origami nanopillar, the capturing strands and the nanoparticles only allowed for the binding of a single nanoparticle (monomer antenna) strongly reducing the achievable enhancement values (average of 7.3).¹⁵ These moderate fluorescence enhancement values were not sufficient for detecting single fluorescence molecules with low-NA optics. For example, our previous work on benchmarking the sensitivity of smartphone-based detection systems suggested that a signal equivalent to at least 16 single emitters is required for detection on a smartphone-based low-NA microscope.¹⁶ Therefore, a diagnostic single-molecule assay fully exploiting the signal amplification potential of DNA origami nanoantennas has not been presented to date and remained highly desirable to enable detection of single molecules with affordable low-NA optics.”

3. The authors have clarified what they meant by "addressable" in their response letter. However, Fig.1 is not really clear. For example, in panel a, the "gap" between the two pillars is only shown in a tiny inset. In panel b, the gap is NOT shown at all. For panel c, it is not clear which feature the readers should focus on. (Lines 64-65 points to panels b and c to illustrate the "gap" between the pillars). Please consider revising this figure.

Response and action: We thank the reviewer for her/his suggestion to improve the representation of our new design. We have revised Figure 1 (panels a-c) to help the reader better visualize the new DNA nanostructure that we report here. Representation of the plasmonic gap, in a fully assembled NACHOS is not straightforward, as bound nanoparticles are “blocking” the front and back view. However, we added a top view representation to Fig. 1b and we also marked each representation of the nanostructure to clarify whether it shows side, top, or front view to avoid confusion. Furthermore, we have also increased contrast in Fig. 1c to make the DNA origami part more visible.

4. There are a lot of errors in the SI:

- a. Supplementary Figure 4 is exactly the same as Supplementary Figure 8, while they are supposed to represent results under completely different conditions.
- b. SI figure 4 caption, "excitation" is misspelled
- c. SI table 5 caption, "supplementary" is misspelled
- d. SI figure 15 is presented in a confusing manner. It seems to be three histograms offset arbitrarily along the horizontal axis?

Response and action: We thank the reviewer for pointing out the errors and the typos. The following changes were made in the manuscript to address them:

- a. We apologize for this mistake. We have now provided the correct updated figure for Supplementary Figure 8.
- b. We corrected the typo in “Excitation”.
- c. We corrected the typo in “Supplementary”.
- d. We updated the Supplementary Figure 15 to show the fluorescence enhancement plots as separate histograms.

5. In Supplementary Note 1, the authors claim that "large scale production of the filters and adapted filter size can reduce the price by at least one order of magnitude." I do not understand where this statement comes from. Please provide more reasoning or source. The current commercial filters are already produced at a large scale. Customized coating runs are still much more expensive (for example, as offered by Chroma).

Response and action: In this statement we did not consider the custom coating, rather the custom size of the filter. Usually commercially available fluorescence filters have a diameter of 25.4 mm, while for example our smartphone microscope requires a filter with a diameter not larger than 4 mm. Therefore, one standard filter could be used to make smaller filters for more than 10 smartphone microscopes. This is the reason why we stated that price can be reduced by one order of magnitude. To clarify this, we made changes in Supplementary Note 1: “Large scale production of the filters with customized size can reduce the price by at least one order of magnitude, as a currently used standard commercially available filter is big enough to provide material for over 10 filters for smartphone microscopes.”

6. Some sentences lack clarity.

- a. Line 80-83 "Single-step photobleaching confirms that the detected signal originates from a single fluorescent molecule with a signal-to-background ratio (SBR) of 7.4 ± 0.9 and 361 ± 35 and a signal-to-noise ratio (SNR) of 4.5 ± 0.2 and 5.6 ± 0.3 for the reference structures and the NACHOS, respectively." this sentence is difficult to read.
- b. Fig 1d caption: "Single-molecule fluorescence intensity transients, measured by confocal microscopy, normalized to the same excitation power of a single Alexa Fluor 647 dye". There is no "normalization" from the figure as count rates were reported.
- c. SI Table 4 caption: "If necessary, some unmodified strands from Supplementary Table 2 and modified strands from Supplementary Table 3 should be left out." I'm not sure what exactly this means.

Response and action: We thank the reviewer for her/his feedback and for the suggestions to improve the clarity of our manuscript. The sentences highlighted by the reviewer above were clarified and addressed as follows:

- a. The sentence in lines 80-83 was changed to: “Single-step photobleaching in the intensity versus time transients (Fig. 1c) confirms that the detected signal originates from a single fluorescent molecule. Further analysis of single-molecule transients demonstrates that the signal-to-background ratio (SBR) could be significantly improved by the nanoantenna (361 ± 35) when compared to the reference structure (7.4 ± 0.9).” We omitted the SNR because it has no explanatory power as both values were measured with different excitation intensity.
- b. In order to clarify what we meant by “normalized to the same excitation power of a single Alexa Fluor 647 dye” we corrected the label of the y-axis in Fig. 1d clarifying that these are the count rates per nW of excitation power (kHz/nW).
- c. The sentence in the caption of Supplementary Table 4, was changed to: “The unmodified staple strands from Supplementary Table 2 and modified staple strands from Supplementary Table 3 which are replaced by the capture strands and should therefore be left out in order to fabricate the NACHOS are indicated in the second column.”

Reviewer #2 (Remarks to the Author):

This revised manuscript is now addressing properly most of the reviewers concerns. I was already quite positive with the early version, and now that it has improved I can recommend its publication in its present form. The additional concerns remaining (limit of detection, NA for detection optics, bioassay application) fall beyond the scope of this paper as they deserve an in-depth study on their own.

Response: We thank the reviewer for the positive feedback on our work and the recommendation to publish it.

Reviewer #3 (Remarks to the Author):

The revised manuscript addresses the majority of comments by the referees appropriately. Although the manuscript still only provides a proof-of-principle sensing assay, I believe that the technological advances made are very convincing and suitable for Nature Communications. It is particularly nice to see that the DNA origami assay is stable in a more realistic biological fluid.

Response: We thank the reviewer for her/his positive reception of our work and the recommendation to publish it in Nature Communications.

I have one final remark regarding line 131, where the authors mention "Incubation of NACHOS with 34-nt long target sequence containing 1-nt, 2-nt and 3-nt mismatches in the target region also demonstrated the selectivity of this assay for small target sequence variations (Supplementary Figure 6)." This can be mis-interpreted as a claim that the assay is very selective and few-nucleotide mismatches can be distinguished. Looking at the results in the SI however it will be difficult to detect mismatches because the binding yield only reduces by 50% for a 3nt mismatch. This is not too surprising since the capture strand contains 17 nucleotide complementary regions that exhibit a sizeable affinity even with 3 mismatches. I don't believe this will be an issue because in the foreseen application it is highly unlikely that strands with only a few mismatches are present at similar concentrations. I do suggest however to rephrase this sentence because it seems to overclaim the results now.

Response and action: We thank the reviewer for her/his comment and suggestion to clarify the claims of our findings. We do agree that the way these results are currently described might lead to a false claim that small nucleotide mismatches can be distinguished and we would like to avoid over claiming our findings. Therefore, the sentence in the manuscript describing these results with different mismatch sequences was changed to: "Incubation of NACHOS with 34-nt long target sequence containing 1-nt, 2-nt and 3-nt mismatches in the target region led to a drop in the number of co-localized spots (Supplementary Figure 6), indicating a certain degree of selectivity in this assay, which potentially can be further improved by optimizing the sequence and length of the DNA capture strand". We believe this is a more precise description of our findings and we thank the reviewer again for his suggestion.